# Development of Open-Assistant Environment for Integrated Operation of 3D Bridge Model and Engineering Document Information

**Sang I. Park [1,2]**, **Bong-Geun Kim [3]**, **Wonhui Goh [4]** and **Goangseup Zi [4,\*]**

1   Department of Civil, Environmental and Architectural Engineering, University of Colorado, Boulder, CO 80309, USA; sang.i.park@colorado.edu
2   Research Institute for Safety Performance, Korea Authority of Land & Infrastructure Safety, Jinju 52856, Korea
3   Taesung SNI Singapore Branch, Singapore 208652, Singapore; bgkim@tssni.com
4   School of Civil, Environmental and Architectural Engineering, Korea University, Seoul 02841, Korea; wonhee0689@korea.ac.kr
\*   Correspondence: g-zi@korea.ac.kr; Tel.: +82-2-3290-3324

**Abstract:** This study proposes a method for assistant environments to integrate 3D bridge model information and engineering document fragments. The engineering document content varies depending on the process. Therefore, we accept a loose coupling concept to support the independence of each information set instead of using a specific data model for effective integration. The engineering document is translated into an Extensible Markup Language (XML)-based structured format based on the explicit and apparent semantic structure of the document. An extended industry foundation classes (IFC) schema is proposed to manage the bridge information model, as well as document fragments. An information document (iMapDoc) is proposed to manage interim data to connect a 3D digital model, an IFC model, and engineering document fragments. Document fragments on a specific component in the 3D bridge model are retrieved to validate the developed integrated assistant module.

**Keywords:** 3D bridge model; IFC-based bridge model; engineering document; document fragment; integrated operation

## 1. Introduction

A certain level of knowledge is required to understand and use construction information presented in the form of engineering documents, such as structural calculation documents. Therefore, an environment that can provide information regarding various types, such as documents or 3D digital models, depending on the intended purpose must be established; this is because completing a construction project is achieved through the collaboration of participants with various knowledge backgrounds. Tatum [1] emphasized the role of a three-dimensional (3D) model-based environment in promoting efficiency in the communication of all architect, engineering, and construction (AEC) project participants, and we agree with his opinion. Building information modeling (BIM) is a specialized application field of 3D model-based integration from an information point of view; furthermore, it is expected to be one of the most important applications for the integrated operation of BIM and document-type information in the AEC field. Data in BIM and documents must be managed, as well as linked or aligned, appropriately to achieve an effective integration of BIM and engineering documents in a system or environment.

The information of model objects is managed internally by the tool itself in the case of closed BIM, which generates a model in a proprietary format using BIM authoring tools. Access to model data is possible within the software or within an application programming interface (API) scope. End-users in the practical field often base their preference on closed BIM from the perspective of information management because model visualization and

object generation/editing are performed within one platform [2]. However, this method may cause reliability issues, owing to restriction to full data access, platform dependency, and non-guaranteed transparency of internal processes [3]. In open BIM represented by industry foundation classes (IFC), access to all model object information and modification is enabled [4] according to the standard data access interface (SDAI) standardized by the International Organization for Standardization (ISO) [5]. This method, however, requires advanced programming knowledge to control the IFC physical file (IPF) directly for modifying geometries, and there are instability issues in the process of importing the IPF from the BIM tools.

No standardized method is available for data access, unlike in BIM, because the content included in the construction engineering document is in a denormalized form; hence, various relevant studies are being conducted. Marking-up on contents can be interpreted by a machine, although semi-automatic or manual preprocessing is required for generating mark-up [6]. BIM-based automated compliance checking (ACC), the core task of which is to identify the semantic meaning of codes or regulations [3], proceeds in a similar manner to that of BIM construction document connection. To identify the semantic meaning from documents, specific rules were assigned to the plain documents to extract the content [7,8], or an ontology model was used [9–11]. Recently, research that extracts meaning from regulations through natural language processing or machine learning has been actively conducted [12–14]. These studies focus on the "latent semantic structure" categorized by Wang et al. [15]. The connection between BIM and regulation focuses on mapping the attributes extracted from each model (BIM and document), rather than document fragments or elements from the ACC perspective [16–18]. This results in relatively high costs for extracting the required information, as well as inefficiency if the documents are used as non-geometric reference information instead of a regulatory review. If engineering documents are used as references, then the approach is effective when the "explicit or apparent semantic structure" is used to reconstruct construction documents [19–21].

Choi et al. [22] has performed BIM-document integration by adding the necessary rules to the IFC entities or properties. Opitz et al. [23] linked document contents stored in a repository with an IFC model via separate link elements. As summarized in Table 1, however, previous research mainly focuses on the perspective of document information extraction, and so do not consider the information connection considering the IFC schema or do not pay much attention to the relationship with the 3D object.

Herein, we studied methods that can link engineering documents as reference information with the bridge information model, and subsequently developed an integrated open-assistant environment. Furthermore, we developed a structuralizing method using the explicit and apparent semantic structure of unstructured plain document contents by improving the work of Kim et al. [20]. We adopted the extended IFC schema proposed by Park et al. [24] for bridge information modeling and described how to manage document information based on IFC. An Autodesk Revit-based add-in module was developed to generate and manage a bridge information model using the adopted IFC schema. An interim mapping document (iMapDoc) generated during the process can serve as an excellent bridge for a seamless interface between the closed and open BIM, as well as providing relevant document fragments for each model component, even when the 3D model objects are modified.

**Table 1.** Overview of related research.

| Previous Study | Extraction of 3D Model Info. | Extraction of Doc. Info. | Integration, Mapping Method | Limitation |
|---|---|---|---|---|
| Hjelseth and Nisbet [6] | X | RASE [1] methodology | Not proposed | Integration method between 3D model and doc. Info. was not proposed. |
| Zhong et al. [9] | X | CQIEOntology [2] (Manual) | Not proposed | Integration method between 3D model and doc. Info. was not proposed. |
| Choi et al. [22] | IFC | X | IFC user-defined property sets (PSETs) | Regulation codes must be mapped into IFC PSETs manually. |
| Opitz et al. [23] | IFC schema | X | SQL and BIMfit Model Query | The document content should already be stored in DB in a fragile state. |
| Beach et al. [7] | IFC | Extended RASE (RASE + XML tag) | Experts performed the mapping between the code fragments and IFC entities. | Mapping was performed manually. |
| Zhou and El-Gohary [10] | IFC | Rule-based OBIE [3] algorithm | IFC–SIE [4]–logic facts transformation | OBIE algorithm highly depends on specific knowledge domain (building energy conservation codes). |
| Sydora and Stroulia [8] | IFC | Rule Language (manual) | IFC-based automatic mapping | The information should be organized to interpret into rule language manually. |

[1] RASE: requirement, applicability, selection, and exceptions. [2] CQIEOntology: quality inspection and evaluation ontology. [3] OBIE: ontology-based information extraction. [4] SIE: semantic information element.

## 2. Document Analysis and Translation to Structured Format

### 2.1. Translation Process from Unstructured Engineering Document to XML Document

An engineering document saved as a plain text file is used as an input file to eliminate errors that may occur when recognizing letters from visualized documents. Figure 1 illustrates the entire process of generating a structured Extensible Markup Language (XML) document from a text document referred to by Kim et al. [20]. As depicted in Figure 1, the input engineering document is translated into an XML document through three main steps, according to the subtitle structure. The first step in performing translation to the structured XML document is to store each sentence of the contents sequentially by classifying strings for the types of heading symbols, headings, subtitles, content, and references into a temporary table based on the engineering document model. The types of heading symbols are used to identify the hierarchy of the contents. The temporary table is rearranged after identifying the sentence structure using the existing data stored in the temporary table. The hierarchical information of the contents is identified using the rearranged data connected to the types of heading symbols and the tree structure of the document. Finally, the XML file is generated using the information saved in the temporary table and hierarchical information.

### 2.2. Content Analysis of Engineering Documents

We defined several notations to describe the content of the engineering documents efficiently, as follows:

1. The string $S = s_1, s_2, \cdots, s_n$ represents a set of characters with a finite sequence; here, $s_k \in \Psi$, $1 \leq k \leq n$, and $\Psi$ is the set of all the characters, including space, in a given document;
2. In $A ::= B$, the symbol ‘$::=$’ implies that $A$ can be expressed as $B$;
3. The symbol ‘$|$’ represents ‘or’.

The text information comprising an engineering document can be separated using finite string sets with sequences; the string set $S^i$ of the $i$th row can be expressed as Equation (1), referring to Kim et al. [20].

$$S^i ::= H^i|C^i|H^iC^i|H^iR^i|C^iR^i|H^iC^iR^i \tag{1}$$

where $H^i$ is a string set for the subtitle, $H^i = s_1, s_2, \cdots, s_l$; $C^i$ is a string set for the document content, $C^i = s_{l+1}, s_{l+2}, \cdots, s_m$, $R_i(= s_{m+1}, s_{m+2}, \cdots, s_n)$ represents the string set for reference, and $0 \le l \le m \le n$.

Figure 2 depicts the content analysis algorithm for extracting the document structure based on Equation (1). More definitions and processes are available in the previous research by Kim et al. [20].

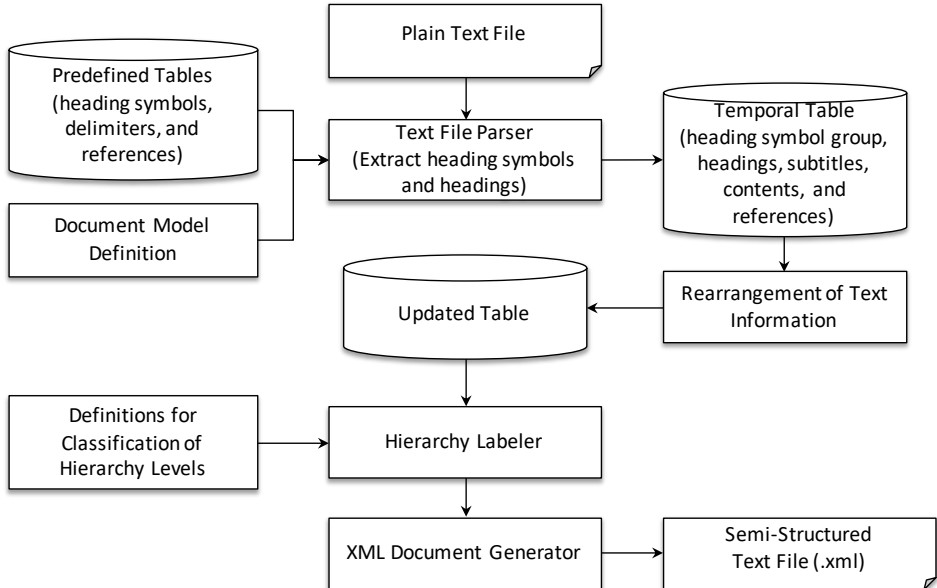

**Figure 1.** Document translation process for text information.

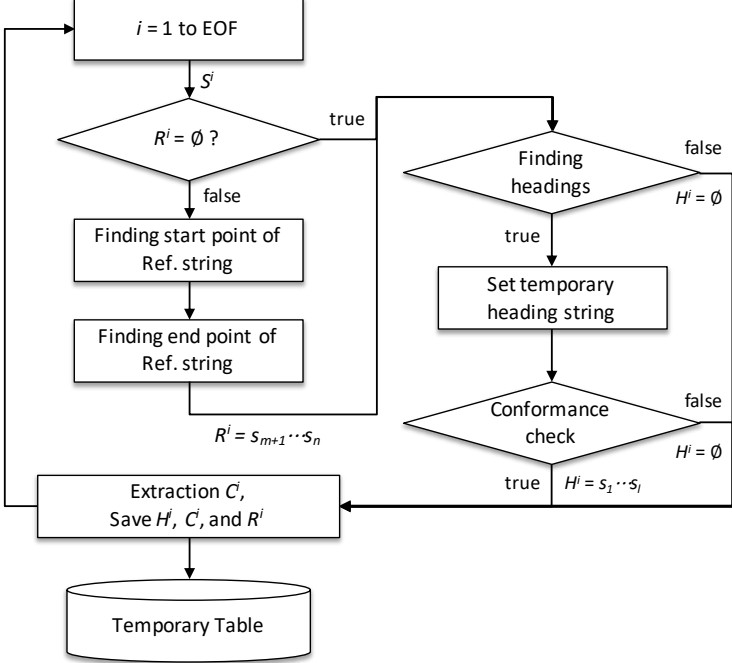

**Figure 2.** Algorithm for content analysis to extract document structure.

### 2.3. Identification of Bullet-Form Text Strings

To identify the syntax's meaning, which comprises the text string of the engineering documents, the temporary table constructed through the component extraction algorithm of the document is used, as described in Section 2.2. $TS^i$, the string set of a random $i$th row, can be expressed as shown in Equation (2).

$$TS^i = \left\{ HS^i_{ID}, hs^i, hc^i, C^i, R^i \right\} \tag{2}$$

where $HS^i_{ID}$ is the unique group number including the heading $hs^i$; $hc^i$ is the string for the title except the heading symbol; and $C^i$ and $R^i$ are text strings for the content and reference, respectively, as defined in Equation (1). We regarded the $i$th row to $i + n$th row of the temporary table contents, when they satisfy all conditional expressions shown in Equation (3a), as the bullet-form text strings. Here, $n = 1, 2, 3, \cdots$.

$$HS^i_{ID} = HS^{i+n}_{ID} \tag{3a}$$

$$hs^{i+\alpha} \in BS^d \quad (\alpha = 0, 1, 2, \cdots \leq n) \tag{3b}$$

$$C^i = \varnothing \wedge C^{i+n} = \varnothing \tag{3c}$$

In Equation (3b), $BS^d$ is the set of heading symbol groups with the depth $d$, where $d$ is a natural number exceeding 1, as defined by users. In Equation (3c), $\varnothing$ means the set with no elements and $\wedge$ logical conjunction or meeting in a lattice. The identified bullet-form text strings according to Equation (3a) are assigned to the $i - 1$th document content ($C$) using Equation (4); subsequently, the contents in the temporary table are rearranged.

$$C^{i-1}_{new} = C^{i-1}_{old} + \sum_{j=1}^{n} \left( nl + hs^j + hc^j \right) \tag{4}$$

where $C^{i-1}_{new}$ is the text string for the newly updated $i - 1$th document content, $C^{i-1}_{old}$ is the string text for the $i - 1$th document content before rearranging the content, and $nl$ is the character that represents a new line.

### 2.4. Identification of Hierarchical Structure of Subtitles

The problem of translating unstructured document contents into a tree-shaped hierarchical structure can be described by estimating the depth of the headings of the corresponding contents. The hierarchical information of the headings is identified by comparing the number of heading symbol groups. We used the results from a previous study that provided a generalized solution to this problem [20]. Using the algorithm described above, the unstructured document content was regenerated into a structured document format (Figure 3) using the developed translator, as shown in Figure 4. The tags of the XML element indicate the content items describing the function of the sentence, and the heading symbols identified for hierarchical classification are expressed as the *header* attributes of the element. If the element should refer to other documents or codes, the references appear in the *reference* attribute of the element. The text data are expressed as parsed character data (PCDATA) of XML. The core functions of the translator shown in Figure 3 are combined into the integrated module to manage the bridge and document model, which will be explained later.

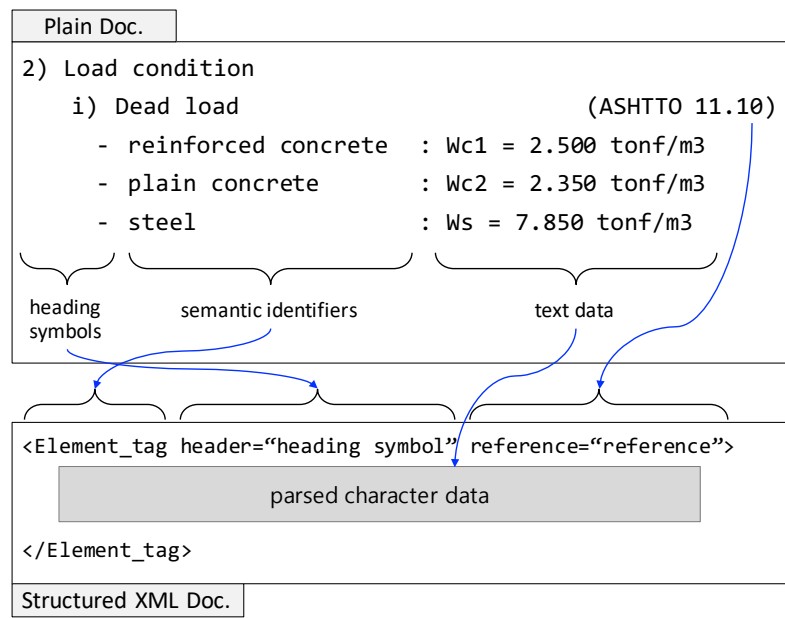

**Figure 3.** Mapping relationship between a plain and XML document.

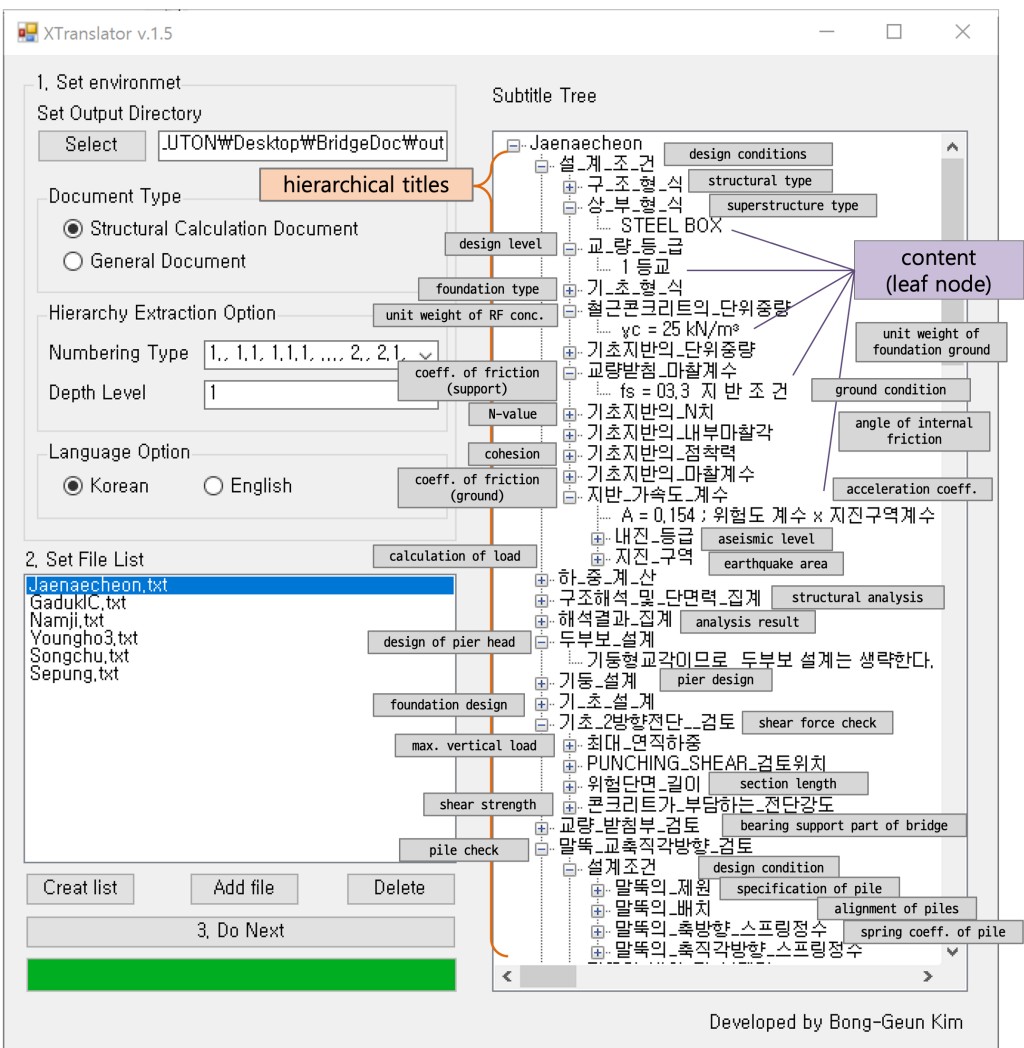

**Figure 4.** Module developed to translate plain document to XML document.

*2.5. Performance Evaluation of Document Translation Module*

　　We used precision and recall, which are widely used in the information retrieval field for a performance evaluation of the component extraction algorithm of a document. Precision and recall use the number of true positive (*TP*), true negative (*TN*), false positive (*FP*), and false negative (*FN*) values; *TP* means that the extracted title sentence is correct, *TN* means that the extracted content sentence is correctly recognized, and *FP* means that the algorithm misrecognizes a sentence as a title. *FN* means that the title sentence is not recognized as the title sentence. The equations used to measure the precision and recall are as follows.

$$Precision = \frac{TP}{TP + FP} \tag{5}$$

$$Recall = \frac{TP}{TP + FN} \tag{6}$$

　　The hierarchies recognized by the proposed algorithm in this study are relative classification among heading symbols; the recognized results of the hierarchical classification of the preceding items affect the following items. Therefore, the module performance of this part was checked following these equations:

$$GAH = \frac{C^G}{TP} \tag{7}$$

$$PAH = \frac{C^P}{TP} \tag{8}$$

where *GAH* means 'generalized accuracy for hierarchy labeler' and *PAH* means 'precise accuracy for hierarchy labeler'. $C^G$ is the number of results from which relative hierarchical classification was performed correctly among TPs, and $C^P$ is the number of results from which absolute hierarchical classification was performed correctly among TPs. Therefore, *GAH* and *PAH* represent ratio values corresponding to $C^G$ and $C^P$, respectively.

　　The proposed algorithm performance, including the content extracting and hierarchical classification processes, was evaluated as the following equations.

$$GAA = \frac{C^G}{TP + FN} \tag{9}$$

$$PAA = \frac{C^P}{TP + FN} \tag{10}$$

where *GAA* means 'generalized accuracy for application module' and *PAA* means 'precise accuracy for application module'.

　　Figure 5 shows the results of applying Equations (5)–(10) to 20 bridge engineering documents with an average number of sentences of 4814 and the number of title sentences to be extracted as 433. The lowest and mean values of PAA are 97.36% and 99.47%, respectively, so it can be judged that the performance as a content item extraction algorithm for BIM-based integration is sufficient.

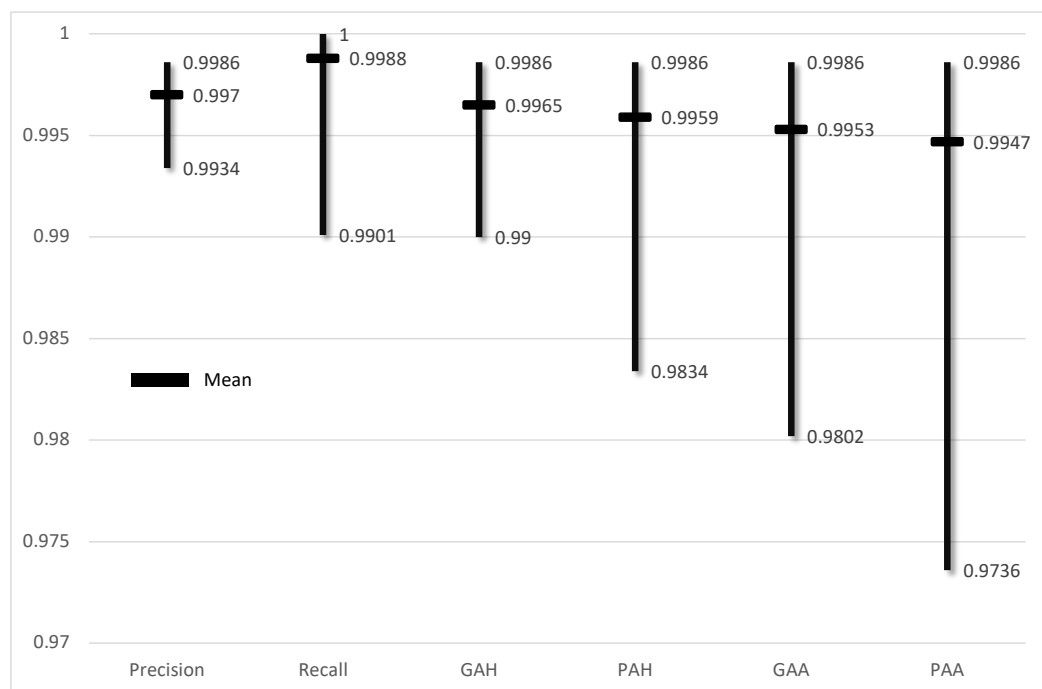

**Figure 5.** Performance evaluations based on precision, recall, GAH, PAH, GAA, and PAA.

### 3. IFC-Based Bridge Information Modeling with Document Metadata

*3.1. IFC Schema for Bridge Model and Document Information*

The most recent official IFC released by buildingSMART International (bSI) is version IFC4.0.2.1 [25]. IFC4.0.2.1 has opened up the possibility of future extensions for civil infrastructures (*IfcCivilElement*), although they will be removed in a future version. The unofficial version currently being developed (IFC4.3RC4) includes alignments (*IfcAlignment*), roads (*IfcRoad*), railways (*IfcRailway*), ports (*IfcMarineFacility*), and bridges (*IfcBridge*). Regarding the bridge structure, IFC4.3RC3 includes the *IfcBridge* entity in order to represent the spatial information of the bridge as a subtype of *IfcFacility*. *IfcBridge* has an enumeration type to embody a spatial function for the subspace of the bridge [26]. More specific spatial components constituting the bridge structure can be managed using the *IfcBridgePartTypeEnum* type of *IfcFacilityPart*. The representation of the physical element of the bridge can be used in the subtypes of *IfcBuiltElement*.

We used the schema developed by Park et al. [24] instead of IFC4.0.2.1 or IFC4.3RC4 for the following reasons:

- IFC4.3RC4 is a schema under development that has not yet been officially released. Therefore, current BIM authoring tools, such as Autodesk Revit, cannot handle the information generated by IFC4.3RC4;
- Information on the bridge structure components covered by IFC4.3RC4 is limited; it does not define bridge-specific and bridge-related attributes that should be treated as entity and attribute level.

*IfcBridgeAddMeshfree*, proposed by Park et al. [24], extends additional entities focusing on the bridge structure and bridge components (see Figure 6). The detailed elements of bridges regarded as enumeration types in IFC4.3RC4 are also defined as entities.

In the *IfcBridgeAddMeshfree* schema, *IfcBridge* is used to manage the spatial information of the bridge structure itself, *IfcBridgeSpan* is used to segregate the spatial information based on the bridge's along the road, and *IfcBridgeSpacePart* is used for the transverse direction of the bridge. In order to represent the physical object of the bridge, it was categorized into girder part, slab part, abutment and pier part, and detailed member component part. Each of these contain enumeration types for detailed types or functions.



External document-related information can be managed through the *IfcDocumentInformation* entity in the IFC schema. The *IfcDocumentInformation* can capture metadata, such as the document name, document purpose, and/or revision information of an external document, but not the document content. We linked each content fragment of a document corresponding to a model component rather than linking an entire document for an entire 3D model because *IfcDocumentInformation* can be associated with all IFC objects through the *IfcRelAssociatesDocument* entity. The implementation of *IfcDocumentInformation*-related information will be described later.

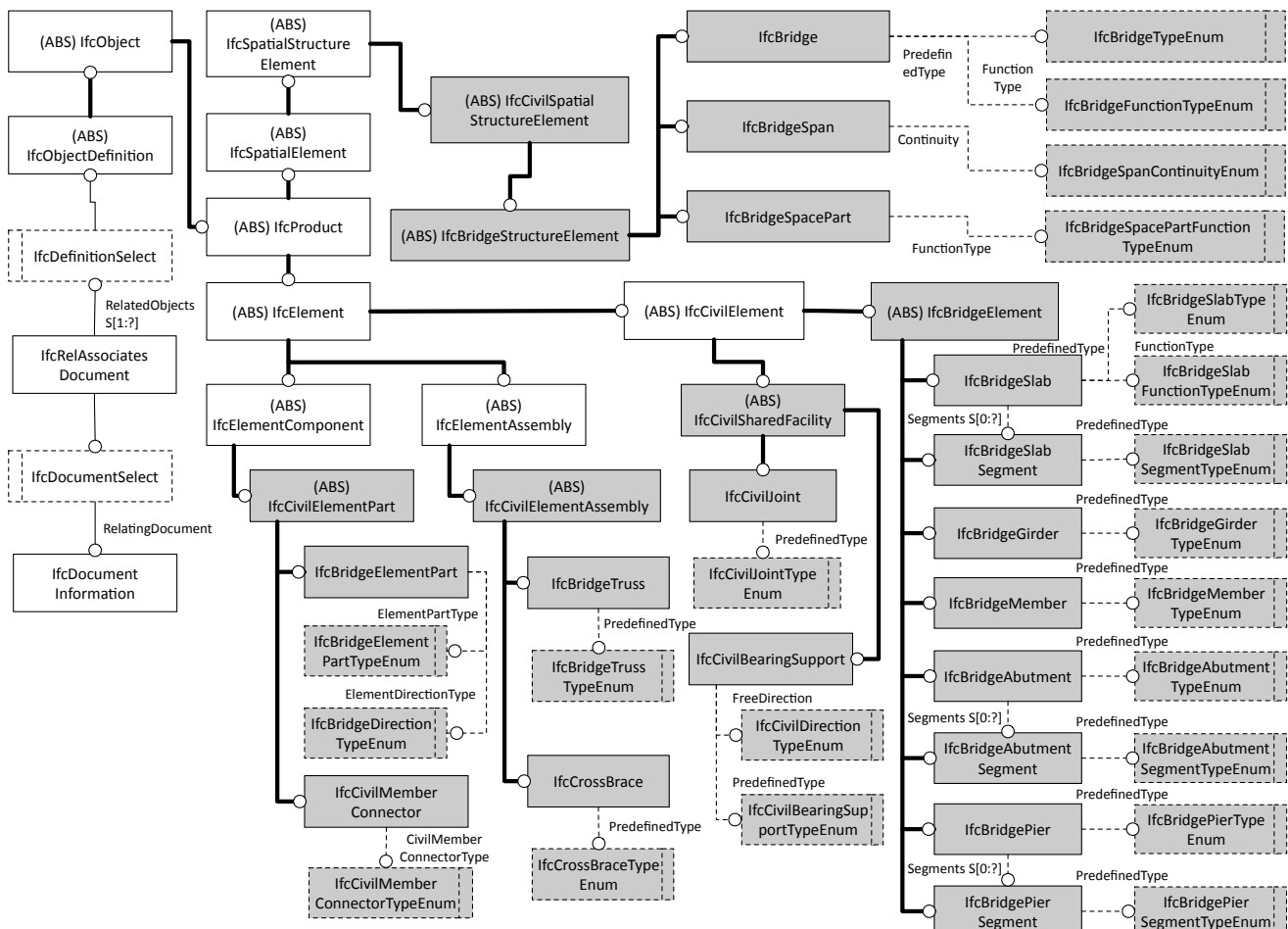

**Figure 6.** EXPRESS-G diagram to manage bridge model and document meta data; the shaded boxes denote extended entities for the bridge structure.

### 3.2. Assistant Module for Integrated Management of Bridge Model and Document Information

We developed a module to manage 3D model information, reflect the extended IFC-based bridge, and enable information retrieval for engineering documents related to bridge segments. The module was based on the Revit API provided by Autodesk [27], a representative BIM authoring tool. This enabled the integration of the extended IFC and engineering document retrieval in the Revit environment. We used the original functions provided by Revit to generate a 3D geometric model (Figure 7a). We modeled the components using general or building elements corresponding to bridge components referred by the concept of Park et al.'s [2] previous research for bridge components, such as bridge piers, that Revit cannot provide. As a note, to connect with the extended IFC entities for the bridge structure, each component should be created as one object. As a preliminary procedure, the user should directly generate spatial objects suitable for bridge and bridge components to apply IFC entities (see Figure 7b). This module suggests an appropriate IFC entity by parsing the EXPRESS files of the extended IFC schema for the spatial object, as depicted

in Figure 7c. All 3D physical objects of the bridge collected automatically through the Revit API can be mapped with an appropriate IFC entity using the interface shown in Figure 7d. The appropriate IFC entity can be suggested through the basic information included in the object, such as the object's family name and description. If the object information has a specific code or words included in the product breakdown structure (PBS) document developed in this study, then the physical object can be mapped precisely with the corresponding IFC entity using this PBS document. The PBS document comprises ab element code (*value* attribute), name (*label* attribute), IFC entity of the extended IFC-based bridge schema (*IfcEntityName* attribute), the IFC4 entity (*Ifc4EntityName* attribute), and the description (*description* attribute). Figure 8 shows a part of the PBS document developed in this study. In this module, the model object can be connected to not only the IFC schema entities developed for bridges, but also to the IFC4 entities, such that other commercial BIM authoring tools can utilize the IFC data generated through this module. Figure 9 shows a part of the iMapDoc containing essential information for the conversion to an extended IFC-based IPF, as well as for connecting a 3D object, IFC, and document fragments. We have designed the iMapDoc based on XML that can generate a relationship between bridge components, as well as connect attributes on the corresponding components. Various tested computer libraries related to the use of XML enable us to skip the verification process of the designed XML schema (iMapDoc).

The *Project* element in line 2 of Figure 9 stores the entire project-related information included in Revit, and is an element that is mapped with the *IfcProject* entity and its attributes of the IFC schema. The child elements of the *StructuralElement* in line 3 are mapped with the physical/spatial entities of the IFC. We used the *IfcSite* entity as the topmost element of all of the spatial elements. The spatial objects generated by the user, shown in Figure 7b, are arranged as *IfcSite*'s sub-entities. The spatial element representing the entire bridge itself is mapped to *IfcBridge* as an extended schema entity and *IfcBuildingStorey* as an IFC4 entity (line 7 of Figure 9). In this case, the *IfcBuilding* entity can be substituted for *IfcBuildingStorey*. Since no entities support bridge structures in IFC4, it is not essential to consider which entities of the IFC4 are used in this study. The entity just needs to be a spatial entity. The spatial objects constituting the entire bridge, such as the bridge section and the upper and lower structures, are mapped with each IFC entity, as shown in lines 11–17 of Figure 9. Line 19 of Figure 9 shows one of the physical elements of the bridge structure, i.e., *Concrete_Pier*, which includes not only the IFC entities (*IfcEntityName*, *Ifc4EntityName*), but also the object number (*OidInSWDB*) managed in Revit and the classification number (*PBSCode*) defined in the PBS. These data can serve as insights for linking information between the Revit objects and the IFC entity software independently, as well as between the IFC entities and engineering documents described in the following sections. The 3D geometries generated in the Revit are represented in the form of boundary representations (B-rep) in the IPF in this study. The vertex and edge information of the B-rep are composed of a separate file, as shown in Figure 10. The iMapDoc can manage this information through the identifier shown in line 24 of Figure 9. The *CompNo* attribute points to the ID of the vertex and triangle data, as shown in Figure 10. The iMapDoc can be converted to the IPF using subtypes of the *IfcRelationship* entity to connect "spatial object–spatial object," "spatial object–physical object," and "physical object–physical object" while sequentially reading elements of the iMapDoc. Property sets (lines 26–33 of Figure 9) in each element are connected to the parent object using the *IfcRelDefinesByProperties* entity in the IFC.

The structure of the document contents can be represented via the tree view, list view, or plain view features provided by the .NET framework, since the engineering document has been translated to the XML format (see Figure 7e). The document content query process is discussed in the next section.

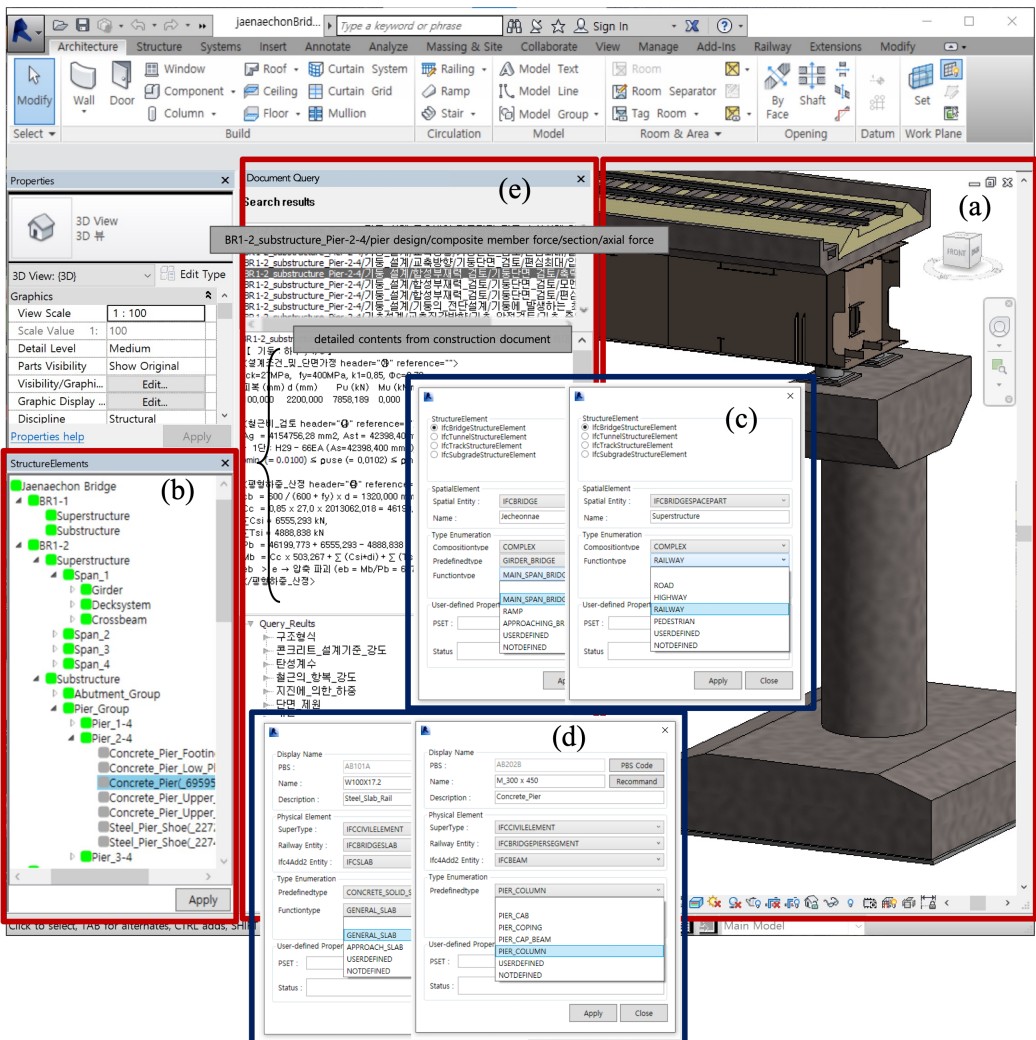

**Figure 7.** Module developed for connecting 3D model, IFC objects, and engineering document: (**a**) 3D bridge model, (**b–d**) user-interfaces for the IFC-based bridge modeling, and (**e**) user-interface for document fragment retrieved based on query.

```
1   <PBS>
2     <StructureElements>
3       ...
4       <Element VALUE="AB" LABEL="Bridge" FILTER="A" IFCENTITYNAME="IfcCivilElementProxy" IFC4ENTITYNAME="IfcBuildingElementProxy"
        DESCRIPTION="It applies to each bridge with respect to roadbed bridges except temporary overpass bridges.">
5         <Element VALUE="AB0" LABEL="General" FILTER="AB" IFCENTITYNAME="IfcCivilElementProxy"
          IFC4ENTITYNAME="IfcBuildingElementProxy" DESCRIPTION="It applies to common facilities of bridges.">
6           <Element VALUE="AB000" LABEL="General" FILTER="AB0" IFCENTITYNAME="IfcCivilElementProxy"
            IFC4ENTITYNAME="IfcBuildingElementProxy" DESCRIPTION="It applies to common facilities of bridges." />
7         </Element>
8         <Element VALUE="AB1" LABEL="Superstructure" FILTER="AB" IFCENTITYNAME="IfcCivilElementProxy"
          IFC4ENTITYNAME="IfcBuildingElementProxy" DESCRIPTION="It applies to the upper structure including the plate and auxiliary
          equipment of the bridge installed above the substructure such as abutments or piers.">
9           <Element VALUE="AB100" LABEL="General" FILTER="AB1" IFCENTITYNAME="IfcCivilElementProxy"
            IFC4ENTITYNAME="IfcBuildingElementProxy" DESCRIPTION="It applies to common facilities of the superstructure of the
            bridge." />
10          <Element VALUE="AB101" LABEL="Plate" FILTER="AB1" IFCENTITYNAME="IfcBridgeSlab" IFC4ENTITYNAME="IfcSlab" DESCRIPTION="It
            applies to the plate of the superstructure of the bridge.">
11            <Element VALUE="AB101A" LABEL="Slab" FILTER="AB101" IFCENTITYNAME="IfcBridgeSlab" IFC4ENTITYNAME="IfcSlab"
              DESCRIPTION="" />
12            ...
13          </Element>
14        </Element>
15      </Element>
16    </StructureElements>
17  </PBS>
```

**Figure 8.** PBS document (shown partially) for mapping between 3D geometry object and IFC entity.

```
1   <MappingIFCRailway4Revit>
2     <Project> <Atts Data="Globalid" ValueType="STRING" Value="JSW9naYtoUSHHv159LOLtQ" /> ... </Project>
3     <StructureElements>
4       <Element Globalid="LDyn4mJBakCAgUWlgS2zMQ" Data="Site" Description="" OidInSWDB="" IfcEntityName="IfcSite"
        isSpatialElement="True" Ifc4EntityName="IfcSite" PBSCode="">
5         <Atts Data="Globalid" ValueType="STRING" Value="LDyn4mJBakCAgUWlgS2zMQ" isEnum="False" />
6         ...
7         <Element Globalid="0+6AkOSlBkGa2k0lqmE9SA" Data="Jaenaechon Bridge" Description="" OidInSWDB="" IfcEntityName="IFCBRIDGE"
          isSpatialElement="True" Ifc4EntityName="IfcBuildingStorey" PBSCode="">
8           <Atts Data="Globalid" ValueType="STRING" Value="0+6AkOSlBkGa2k0lqmE9SA" isEnum="False" />
9           ...
10          <Atts Data="Functiontype" ValueType="IFCBRIDGEFUNCTIONTYPEENUM" Value="" isEnum="True" EnumData="MAIN_SPAN_BRIDGE" />
11          <Element Globalid="OOyLl1P2y0GOtXVgrpyrrA" Data="BR1-2" Description="" OidInSWDB="" IfcEntityName="IFCBRIDGESPAN"
            isSpatialElement="True" Ifc4EntityName="IfcBuilding" PBSCode="">
12            ...
13            <Element Globalid="xmPXvzodgkK4b3qwdcgK/g" Data="Substructure" Description="" OidInSWDB=""
              IfcEntityName="IFCBRIDGESPACEPART" isSpatialElement="True" Ifc4EntityName="IfcBuildingStorey" PBSCode="">
14              ...
15              <Element Globalid="pC8mMNk8q0qI3DqQRVB1MQ" Data="Pier_Group" Description="" OidInSWDB=""
                IfcEntityName="IFCBRIDGESPACEPART" isSpatialElement="True" Ifc4EntityName="IfcBuildingStorey" PBSCode="">
16                ...
17                <Element Globalid="Ob/QaX0wv0Gnwtei0/RA6A" Data="Pier_2-4" Description="" OidInSWDB=""
                  IfcEntityName="IFCBRIDGESPACEPART" isSpatialElement="True" Ifc4EntityName="IfcBuildingStorey" PBSCode="">
18                  ...
19                  <Element Globalid="pSMyBLhGB06HLAvQv7x8RA" Data="" Description="Concrete_Pier" OidInSWDB="695957"
                    IfcEntityName="IFCBRIDGEPIER" isSpatialElement="False" Ifc4EntityName="IFCCOLUMN" PBSCode="AB202">
20                    ...
21                    <Atts Data="Predefinedtype" ValueType="IFCBRIDGEPIERTYPEENUM" Value="" isEnum="True" EnumData="T_TYPE" />
22                    <Atts Data="Segments" ValueType="SETIFCBRIDGEPIERSEGMENT" Value="" isEnum="False" />
23    [pier_design]   <Doc xpathEle="/기둥_설계" xpathAtt="header" xpathAttVal="9." />
24                    <Geometry> <Vertices Name="M_300 x 450" CompNo="327_0" /> <Triangles Name="M_300 x 450" CompNo="327_0" />
25                    </Geometry>
26                    <PSETList>
27                      ...
28                      <PSET Globalid="ppJgvWJtvUKlYQauq9XKxg" Name="Dimensions">
29                        <Property Name="Area" Value="600.256173864946" />
30                        <Property Name="Length" Value="39.3700787401575" />
31                        ...
32                      </PSET>
33                    </PSETList>
34                    ...
```

**Figure 9.** iMapDoc (shown partially) connecting 3D model object, IFC data, and engineering document fragment.

```
1   <Geometry>
2       ...
3       <Vertices CompNo="327_0">
4           -98.735870406,3.922485127,0.000000000
5           -98.608448124,3.916225271,0.000000000
6           -98.482252987,3.897505991,0.000000000
7           -98.358500326,3.866507563,0.000000000
8           -98.238381944,3.823528519,0.000000000
9           ...
10      </Vertices>
11      <Triangles CompNo="327_0">
12          18,20,21
13          22,23,21
14          18,21,23
15          15,16,17
16          19,20,18
17          ...
18      </Triangles>
19      ...
20  </Geometry>
```

**Figure 10.** Geometric information to represent the 3D object in IPF.

## 4. Experimental Verification via Retrieval of Document Fragments Related to the 3D Model Object

*4.1. Process of the Experimental Verification for Connected Document Fragments and 3D Model Object*

The contents in the XML-formatted structured engineering document can be retrieved using the information included in the selected object of the bridge components via the interface shown in Figure 7a or Figure 7b. XML element tags, attributes of the element, and text data (PCDATA of an XML element) are used as keywords to match a specific segment in the document fragments. The query, response, and management of the XML data process are performed by combining the document object model (DOM) to treat XML data for language independently adopted as standard by W3C [28], regular expression to search string patterns introduced by Kleene [29], and XPath defined by W3C as a standard XML data query language [30]. Figure 11 shows the conceptual process for retrieving document fragments from structured engineering documents. The entire document retrieval process occurs in the Revit environment, as shown in Figure 11. Information on the 3D model can use both the iMapDoc developed according to the process described in Section 3.2 and the data included in the IPF. Figure 12 illustrates the process that occurs during the *Query process* shown in Figure 11. Exact word matching is prioritized; however, if an exact matching element does not exist, then the module matches semantically similar words using the *Dictionary for synonyms* shown in Figure 11. The *Dictionary for synonyms* database was developed in this study to increase the accuracy of word matching; it comprises the attributes of *ID*, *Korean Word*, *English Word*, *Abbreviation*, *Symbol*, and *Synonym*.

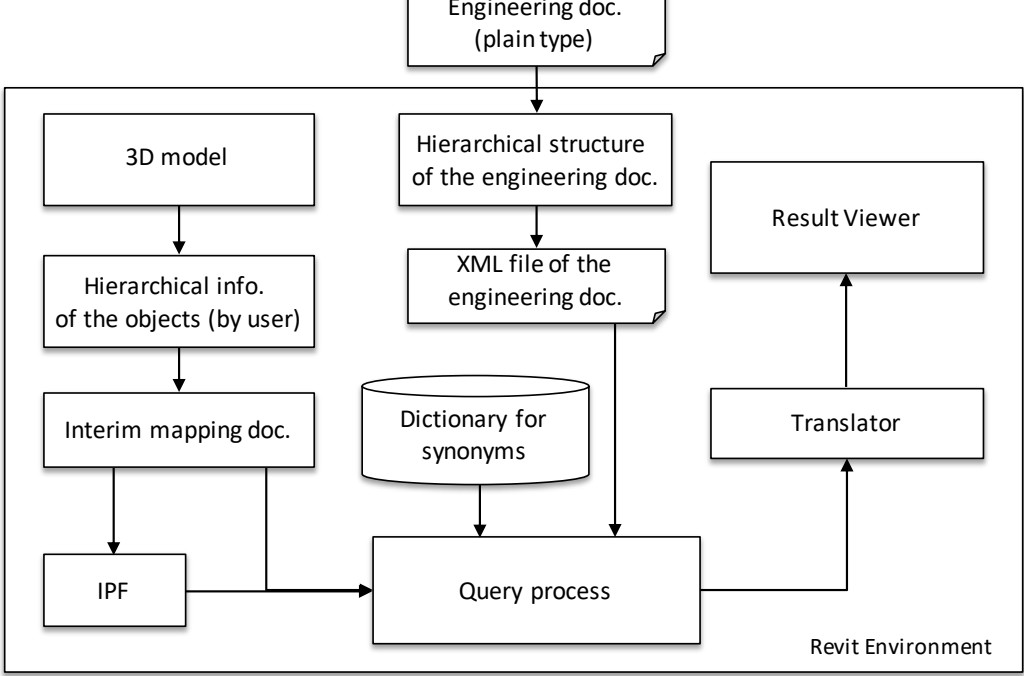

**Figure 11.** Process for retrieving document fragments from structured document in 3D model view.

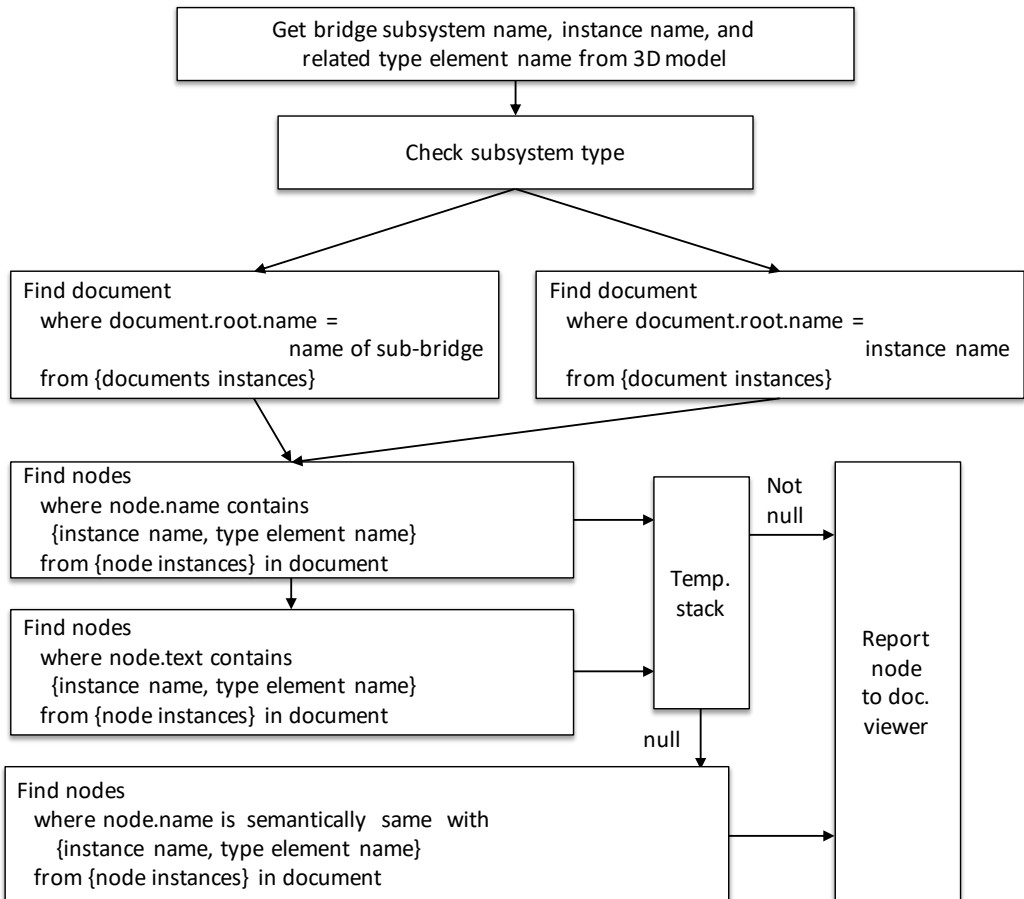

**Figure 12.** Query process for extracting specific nodes from structured engineering document in 3D model view.

*4.2. Connection Review of the Retrieved Document Fragments and the 3D Model Objects*

Figure 13 shows a part of the data of the retrieved structured engineering document related to the *Concrete_Pier* object under *Pier_2-4*. These data are presented in Figure 7e, where the results are shown in a list view (① of Figure 14). *Pier_2-4*, as well as its parent and children elements, can be used to retrieve *Concrete_Pier*-related elements and information because the object selected in Figure 7b is synchronized with iMapDoc (see line 19 of Figure 9 and ② of Figure 14). The *Dictionary for synonyms* database was also used to retrieve the related content. The results selected by the user among the retrieved document contents are integrated with the IPF. The connection information between the 3D object and document content is stored in the *Doc* element in iMapDoc, as shown in line 23 of Figure 9 (③ of Figure 14). The connection information indicates the location of the related document content translated into XML format according to the method described in Section 2 using XPath; line 23 of Figure 9 represents $./pier\_design[@header = 9.]$. The *Doc* element can be added to all children elements of *StructureElements* of iMapDoc. The selected object from the 3D model or iMapDoc can be connected to the IFC entity mutually through the element name and the *GlobalId* attribute of the iMapDoc. Figure 15 shows a part of the IPF generated using a developed converting module including the *Pier_2-4* object. The *Pier_2-4* object was implemented as the *IfcBridgeSpacePart* entity, and its attributes are consistent with the attributes of *GlobalId* and *Data* in line 17 of Figure 9. *Concrete_Pier* was implemented as the *IfcBridgePier* entity, as shown in *#2976952* in Figure 15, corresponding to line 19 of Figure 9 (④ of Figure 14). The elements of iMapDoc and the entities of IPF share the entity name and global ID; the IFC and Revit are connected through the information of *OidInSWDB* and *PBSCode* stored in iMapDoc. The *Representation* attribute representing the shape information of the *IfcBridgePier* entity uses a subtype of the *IfcProductRepresentation*

entity, and the *IfcProductDefinitionShape* entity is used for the representation, as shown in #1273678. The actual data are implemented using the *IfcShapeRepresentation* entity (#1274058), which utilizes the information in line 24 of Figure 9 (⑤ of Figure 14). The *IfcBridgeSpacePart* (#2971755) and *IfcBridgePier* (#2976952) were connected through the *IfcRelContainedInSpatialStructure* entity (#1268533) since they are spatial and physical entities defined by the extended IFC, respectively. The *pier_design*-related information (line 23 of Figure 9) of the engineering document connected to the object of *Concrete_Pier* (line 19 of Figure 9) represents the *IfcDocumentInformation* entity (shown in #4852759 in Figure 15). Among the 17 attributes of the *IfcDocumentInformation* entity in the IFC4 schema, we generated values for attributes *Identification*, *Name*, *Description*, *Location*, *ElectronicFormat*, and *Status* automatically based on iMapDoc. The *Identification* attribute is used to identify documents, and the value data (*pier_design-pSMyBLhGB06HLAvQv7x8RA*) combined with the name (*pier_design*) of the connected document fragment and the *GlobalId* attribute data (*pSMyBLhGB06HLAvQv7x8RA*) of the parent element (*Concrete_Pier*) are used (⑥ of Figure 14). The document fragment name (*pier_design*) is used for the *Name* attribute, and the file name of the XML-formatted construction document (*Jaenaechon.xml*) is represented in the *Description* attribute. The *Location* attribute specifies the location of the document in the form of a uniform resource identifier (URI); it ($Jaenaechon\,/\,pier\_design\,[@header = 9.]$) is represented as using XPath, combining the path of the root element (*Jaenaechon*) of the construction document and the attribute data in the *Doc* element in Figure 9 to specify a concrete path for querying the document fragment. We specify *application/xml* according to the XML Multipurpose Internet Mail Extension (MIME-type) as the attribute data of *ElectronicFormat* for the type of media. The *Status* attribute indicates the document's status represented by the *IfcDocumentStatusEnum* type, and we designated the *.FINAL.* value by default.

This example verifies that 3D models, engineering documents, and IFC can be run together using the proposed integrated approach. In particular, the document files were not deliberately reconstructed and were used as they were created in the bridge design work. Furthermore, we can map information objects from different sets of information using the names of the sub-bridges and components. The same names for subsystems and bridge components are typically used in bridge design drawings and documents. Therefore, it is believed that this method can facilitate the establishment of an integrated information environment for 3D bridge models and engineering documents.

**Figure 13.** Data (shown partially) of retrieved structured engineering document related to *Concrete_Pier* object.

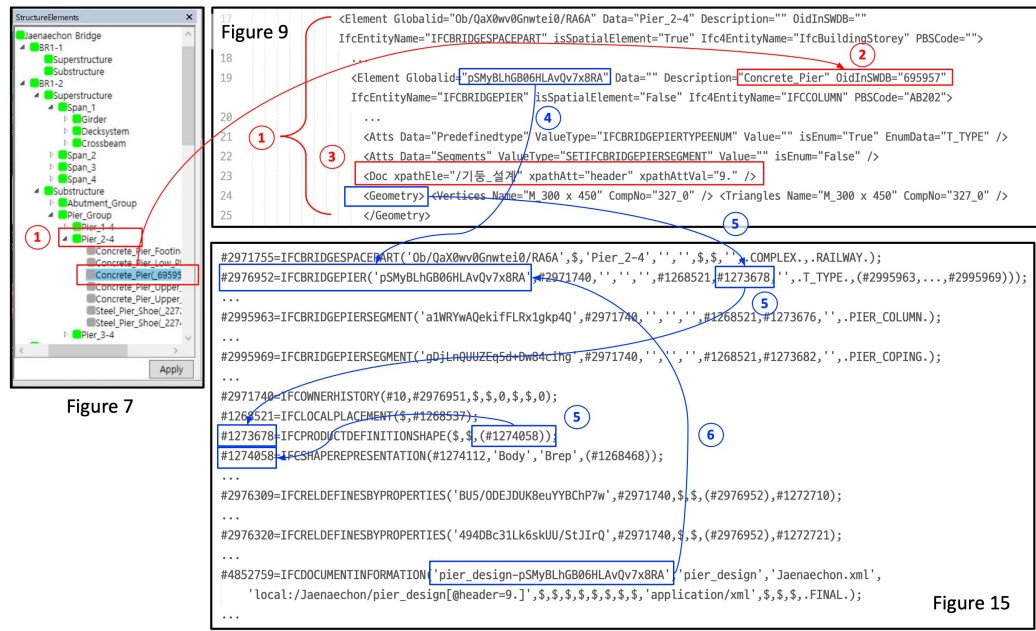

**Figure 14.** Information flow among BIM authoring tool–IFC–engineering document.

```
...
#1268533=IFCRELCONTAINEDINSPATIALSTRUCTURE('iOrfjRtaGEep7VgmZGFXSw',#2971740,'Pier_2-4',$,(...,#2976952,...),#2971755);
#1348174=IFCRELASSOCIATESDOCUMENT('Q/GRFDkNSGK5fo/gWpcRiw',$,$,$,(#2976952),#4852759);
...
#2971755=IFCBRIDGESPACEPART('Ob/QaX0wv0Gnwtei0/RA6A',$,'Pier_2-4','','',$,$,'',.COMPLEX.,.RAILWAY.);
#2976952=IFCBRIDGEPIER('pSMyBLhGB06HLAvQv7x8RA',#2971740,'','','',#1268521,#1273678,'',.T_TYPE.,(#2995963,...,#2995969)));
...
#2995963=IFCBRIDGEPIERSEGMENT('a1WRYwAQekifFLRx1gkp4Q',#2971740,'','','',#1268521,#1273676,'',.PIER_COLUMN.);
...
#2995969=IFCBRIDGEPIERSEGMENT('gDjLnQUUZEq5d+Dw84cihg',#2971740,'','','',#1268521,#1273682,'',.PIER_COPING.);
...
#2971740=IFCOWNERHISTORY(#10,#2976951,$,$,0,$,$,0);
#1268521=IFCLOCALPLACEMENT($,#1268537);
#1273678=IFCPRODUCTDEFINITIONSHAPE($,$,(#1274058));
#1274058=IFCSHAPEREPRESENTATION(#1274112,'Body','Brep',(#1268468));
...
#2976309=IFCRELDEFINESBYPROPERTIES('BU5/ODEJDUK8euYYBChP7w',#2971740,$,$,(#2976952),#1272710);
...
#2976320=IFCRELDEFINESBYPROPERTIES('494DBc31Lk6skUU/StJIrQ',#2971740,$,$,(#2976952),#1272721);
...
#4852759=IFCDOCUMENTINFORMATION('pier_design-pSMyBLhGB06HLAvQv7x8RA','pier_design','Jaenaechon.xml',
    'local:/Jaenaechon/pier_design[@header=9.]',$,$,$,$,$,$,$,'application/xml',$,$,$,.FINAL.);
...
```

**Figure 15.** IPF (shown partially) generated using integrated assistant module related to *Pier_2-4* object.

## 5. Conclusions

Providing appropriate information in a valid and accessible format to construction participants with various knowledge backgrounds is the most important consideration for successful construction and management activities. Research pertaining to the efficient interconnectivity between 3D models and engineering records stored in documents is still in its infancy, whereas research for providing pertinent information based on the visibility of 3D model applications has been actively conducted in recent decades. As one of the methods to effectively deliver information used in the construction and management of bridge structures to users, we focused on integrating the bridge information model and engineering documents, which serve as references throughout the lifecycle. To achieve this, we first proposed a structuralizing method to transform unstructured plain text-typed engineering documents to XML documents by classifying titles and contents, defining hierarchies, and reconstructing them using explicit and apparent semantic structures. Second, an extended IFC schema that can handle bridge structure information was adopted, and IFC entities that

can connect IFC and document information were selected. Finally, a Revit-based add-in module was developed to assist in the integrated operation of bridge information models and engineering documents. The module includes the functions of generating IFC spatial objects, placing physical objects into spatial objects, and interconnecting XML document contents related to model objects. Furthermore, it generates an extended IFC-based IPF that retains hierarchical object relationships and document connection information.

The main contribution of this study was the proposal of a new approach toward achieving an interconnected and integrated operation of the BIM authoring tool–IFC–engineering document using an interim information document named iMapDoc beyond the IFC–engineering document linkage. Each specialized engineering process evolves continuously as new engineering techniques and design philosophies are developed, which can naturally support the process conducted in each independent engineering domain. We expect the proposed process in this study to serve as a reference for future studies as follows:

- Document content management using the IFC: Although the IFC contains most of the information that needs to be dealt with during the entire lifecycle of a facility, few studies suggest a technical process to control the document fragments. This study explained how IFC manages document fragments with examples;
- Smart infrastructure: The successful adaptation of BIM for buildings promotes the growth of BIM for infrastructure for operating smart infrastructure. Interoperability, as well as data mapping between physical and digital models, are considered to be some of the essential keys of successful BIM for infrastructure [31,32]. The integrated operation process of the BIM authoring tool–IFC–engineering document proposed in this study can be a good reference.

Several issues should be addressed to implement the proposed integrated approach more comprehensively in practical working environments. The proposed approach uses the names of instances and type objects (e.g., types of cross-section, material, and connection) to map different information sets. Therefore, a systematic naming rule should be established to identify bridge components and object types based on the consensus of the project participants. Another drawback is that the developed module cannot process contents in figures or tables. However, the text information contained in the tables can be handled using a specific element node in a structured document. This indicates that the figures and tables can be correctly provided with subtitles.

**Author Contributions:** Conceptualization, S.I.P., B.-G.K. and G.Z.; methodology, S.I.P.; software, S.I.P. and B.-G.K.; validation, S.I.P. and W.G.; formal analysis, S.I.P.; investigation, S.I.P. and B.-G.K.; resources, S.I.P. and B.-G.K.; data curation, S.I.P. and W.G.; writing—original draft preparation, S.I.P. and B.-G.K.; writing—review and editing, S.I.P. and G.Z.; visualization, S.I.P. and B.-G.K.; supervision, G.Z.; project administration, G.Z.; funding acquisition, G.Z. All authors have read and agreed to the published version of the manuscript.

**Funding:** This work was partially supported by the National Research Foundation of Korea (NRF) grant funded by the Korea government (MSIT; No. NRF-2021R1A5A1032433).

**Conflicts of Interest:** The authors declare no conflict of interest.

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
