# Peer review of "Development of Open-Assistant Environment for Integrated Operation of 3D Bridge Model and Engineering Document Information"

_applsci, doi:10.3390/app12052510_

Round 1
Reviewer 1 Report
The paper presents an interesting subject, the BIM application is very useful in bridge monitoring, but in the following some comments will be listed in order to improve the paper:
- The references are not shown, so the originality of the paper, considering the absence of state of the art, is not properly demonstrated.
- The validation of the proposed procedure and code is not fully explored.
- The figures could be improved.
Author Response
Response to Reviewer #1:
C1. The references are not shown, so the originality of the paper, considering the absence of state of the art, is not properly demonstrated.
R1. We have described previous studies related to ontology model, Natural Language Processing (NLP), or Machine Learning (ML) for integration or interface of BIM and construction documents in Chapter 1 Introduction. We have also reviewed the connection between BIM and regulations using IFC or external repository from the Automated Compliance Checking (ACC) point of view.
To further clarify this issue, we added a table 1 on Chapter 1, the following sentences are included in 69th – 72nd lines on page 2: “As summarized in Table 1, however, previous researches mainly focus on the perspective of document information extraction, so they do not consider the information connection considering the IFC schema or do not pay much attention to the relationship with the 3D object.”
C2. The validation of the proposed procedure and code is not fully explored.
R2. 1) This study's essential objective is to propose an integration of the document content and the 3D bridge model. We applied precision and recall as well as generalized accuracy for hierarchy labeler (GAH), precise accuracy for hierarchy labeler (PAH), generalized accuracy for application module (GAA), and precise accuracy for application module (PAA) proposed in this study to evaluate the performance of the developed algorithm and module for document analysis.
To further clarify this issue, we added the section 2.5 on pages 7 – 8 of the manuscript.
2) 3D bridge model utilizes BIM based on the IFC schema, and the methodology to extend and utilize the IFC has already been done through previous studies. Section 3.1 describes the IFC and its extension. An important way to connect document content and bridge information model is to use the iMapDoc shown in Figure 9. The iMapDoc was designed by XML for guarantees clarity of the use of the code and process.
To further clarify this issue, the following sentences are included in 248th – 252nd lines on page 11, section 3.2 of the manuscript: “We have designed the iMapDoc based on XML that can generate a relationship between bridge components, as well as connect attributes on the corresponding components. Various tested computer libraries related to the use of XML make skipping the verification process of the designed XML schema (iMapDoc).”
C3. The figures could be improved.
R3. We have improved Figures 8 – 10, 13, and 14.
Reviewer 2 Report
The paper “Development of open-assistant environment for integrated operation of 3D bridge model and engineering document information” needs the following revisions, please.
“Because the document contents change with the corresponding engineering process”: rephrase, please. This is not clear.
There are a lot of [?]. Please find a way to avoid this.
English usage: to improve.
“reflect the extended IFC-based bridge”: what do you mean?
Where are the references?
Is the content original? why? where (which parts?)
Management. What about the need to incorporate sensor-based information? is there any possibility to deal with smart roads (cf. 10.3390/electronics8101180 and 10.1016/j.autcon.2018.07.001).
Overall. the paper deals with two different scientific fields: civil engineering and information engineering. the level of clarity appears unsatisfactory.
Author Response
Response to Reviewer #2:
C1. Because the document contents change with the corresponding engineering process”: rephrase, please. This is not clear.
R1. The abstract was rewritten as follows: This study proposes a method for assistant environments to integrate 3D bridge model information and engineering document fragments. The engineering document content varies depending on the process. Therefore, we accept a loose coupling concept to support the independence of each information set instead of using a specific data model for effective integration.
C2. There are a lot of [?]. Please find a way to avoid this. Where are the references?
R2. It seems that an unknown problem took into place when the submission was forwarded to the reviewers. There is no [?] in our submission. All the references were cited correctly. Given that you pointed out ‘?’ symbol in Figure 5 of the revised manuscript, Figure 5 shows data modeling using the EXPRESS-G method. In EXPRESS-G, [a:b] represents the cardinality, and ‘a’ stands for a minimum value of the attribute, and ‘b’ stands for maximum value, where ‘?’ means for unbound condition. The meaning of the symbols used in EXPRESS-G was omitted in this manuscript.
C3. “reflect the extended IFC-based bridge”: what do you mean?
R3. As specified in section 3.1, the current BIM authoring tools can only represent elements defined in the officially released IFC4, and IFC4 does not include representing the bridge structures. Figure 6 shows the additional entities to represent the bridge structure based on the IFC architecture. The extended IFC means the IFC with additional components to represent the facility. The extended IFC, however, cannot be dealt with in the current commercial BIM authoring tools. Therefore, it needs to develop a new module or add-in package to reflect the extended data schema, IFC, to manage the extended IFC-based information model. Figure 7 shows the developed module to manipulate the extended IFC-based information model and related document fragments.
C4. Is the content original? why? where (which parts?)
R4. Chapter 2 and section 3.2 are about the new method presented in this manuscript. Chapter 2 shows the process of extracting document fragments by identifying the structure of the document, and section 3.2 shows the method of connecting the document model and IFC-based BIM.
C5. Management. What about the need to incorporate sensor-based information? is there any possibility to deal with smart roads (cf. 10.3390/electronics8101180 and 10.1016/j.autcon.2018.07.001).
R5. 1) Although parts of IFC deal with sensor and sensor-based information, this manuscript deals with the connection between document information and IFC schema. It is different from managing sensor-based acquired data compared with the document content, but we think the proposed method in this manuscript can be a reference.
2) We agree that there is a high possibility of linking this research and studies on smart road in future research. In 10.1016/j.autcon.2018.07.001, four papers published by the current authors were cited in the fields of “IFC-based Bridge Information Modeling (BrIM),” “Interoperability,” “Sustainability,” “data management,” and “quantity take-off applications”. We proposed the use of open BIM standard and IFC data schema in those four papers. We have a plan to research open BIM applications to smart roads in future.
Reviewer 3 Report
The presented manuscript has no scientific value as for the article. The presented information may be useful, but more to the technicians than the scientific community. The authors presented the IT tool for the preparation of drawings. The manuscript is not suitable to publish in a scientific journal.
Nevertheless, the manuscript does not contain the literature, the text is very hard to read, conclusions are obvious without any recommendations to the readers.
Author Response
C1. The presented manuscript has no scientific value as for the article. The presented information may be useful, but more to the technicians than the scientific community. The authors presented the IT tool for the preparation of drawings. The manuscript is not suitable to publish in a scientific journal.
Nevertheless, the manuscript does not contain the literature, the text is very hard to read, conclusions are obvious without any recommendations to the readers.
R1. This study does not show a connecting document content fragments and a 3D information model based on a specific case but proposes a process of information connection which is not dependent on a specific environment. Chapter 2 and section 3.2 are about the new method presented in this manuscript. Chapter 2 shows the process of extracting document fragments by identifying the structure of the document, and section 3.2 shows the method of connecting the document model and IFC-based BIM.
We applied precision and recall as well as generalized accuracy for hierarchy labeler (GAH), precise accuracy for hierarchy labeler (PAH), generalized accuracy for application module (GAA), and precise accuracy for application module (PAA) proposed in this study to evaluate the performance of the developed algorithm and module for document analysis. Chapter 4 confirms the originality of the method presented in this study through the check of detailed data.
Reviewer 4 Report
The paper presents an interesting topic; however, the paper is not ready for the publication yet. Engineering documentation system for bridge construction and inspection is always a challenge for all bridge owners. Several associations and/or owners have tried to build up a harmonized code for the bridge/product breakdown structures (PBS) as per provided through several US Department of Transportations (DOT), AASHTO / FHWA, the Ministry of Transportation of Ontario (MTO) in Canada, and ASTM E2103/E2103M-19 Standard Classification for Bridge Elements – Uniformat II. Documentation herein to include project specifications in plain text, design sheets, drawings, and site photos.
The proposed Revit-based add-in Industry Foundation Classes (IFC) based bridge model didn’t investigate the existing documentation system through various DOTs, and how they built the file name convention system, filing formats (i.e. XLS, DOC, DWG, JPG, PDF, etc.) and filing system that helps them to retrieve the data through a semi-automatic process. Still the developed iMapDoc beyond the IFC is not clear in how any report can be generated and how to see several document versions for the same bridge element. Not quite sure how different file types can be converted into XML with plain text.
On the other hand, the paper has missed all citation, replace ? with actual citation, and has no section for the references. Some equations as 3c and its symbols needs to have some explanation in the text. Figure 5 as discussed earlier has to align with the DOT/AASHTO/FHWA requirements in creating the tree and its children.
Author Response
Response to Reviewer #4:
C1. The proposed Revit-based add-in Industry Foundation Classes (IFC) based bridge model didn’t investigate the existing documentation system through various DOTs, and how they built the file name convention system, filing formats (i.e. XLS, DOC, DWG, JPG, PDF, etc.) and filing system that helps them to retrieve the data through a semi-automatic process. Still the developed iMapDoc beyond the IFC is not clear in how any report can be generated and how to see several document versions for the same bridge element. Not quite sure how different file types can be converted into XML with plain text.
R1. This study proposes a method to interface or integrate generated engineering document content fragment and IFC-based BIM separately. The design document was used to review the applicability of the proposed method. In the process of the document analysis process, the format or type of the document is discussed in previous studies (10.1061/(asce)cp.1943-5487.0000047, reference no. 20) and is out of scope in this study.
C2. On the other hand, the paper has missed all citation, replace ? with actual citation, and has no section for the references.
R2. It seems that an unknown problem took into place when the submission was forwarded to the reviewers. There is no [?] in our submission. All the references were cited correctly. Given that you pointed out ‘?’ symbol in Figure 5 of the revised manuscript, Figure 5 shows data modeling using the EXPRESS-G method. In EXPRESS-G, [a:b] represents the cardinality, and ‘a’ stands for a minimum value of the attribute, and ‘b’ stands for maximum value, where ‘?’ means for unbound condition. The meaning of the symbols used in EXPRESS-G was omitted in this manuscript.
C3. Some equations as 3c and its symbols needs to have some explanation in the text.
R3. In Eq. (3c), ∅ means the set with no elements. ∧ means logical conjunction or meet in a lattice.
To further clarify this issue, the following sentence is added in 131st – 132nd lines on page 5, section 2.3: “In equation (3c), ∅ means the set with no elements and ∧ logical conjunction or meet in a lattice.”
C4. Figure 5 as discussed earlier has to align with the DOT/AASHTO/FHWA requirements in creating the tree and its children.
R4. Understandably, ISO 12006-2 based classification systems such as Uniclass and OmniClass are very similar to the IFC schema. The difference is whether it has attributes or not. But the uses and purposes of the classification system and data schema are different. I think that the classification system and IFC schema should be connected, not included. BuildingSMART, an organization that develops the IFC schema, also seems to agree with this point of view. IFC already defines IfcClassification, IfcClassificationReference, and IfcRelAssociatesClassification entities to connect with the elements of the classification system, not to merge.
The extended IFC for bridge structures shown in Figure 5 shows only the part where entities are added to the existing current IFC, and I think that aligning with other classification systems is out of the scope of this study.
Nevertheless, as shown in Figure 8 of the revised manuscript, we still reflect the importance of linking PBS and IFC by suggesting and utilizing an XML-based document including PBS information that can flexibly map PBS and IFC entities.
Round 2
Reviewer 2 Report
The authors carried out several revisions and improved the paper. Although the "accept in the present form" is here selected, it is again noted that a brief discussion of the potential application to the new frontier of smart roads and real-time decision platforms would add a lot to the manuscript (cf. C5. Management. What about the need to incorporate sensor-based information? is there any possibility to deal with smart roads (cf. 10.3390/electronics8101180 and 10.1016/j.autcon.2018.07.001)).
Kind Regards
Author Response
C1. The authors carried out several revisions and improved the paper. Although the "accept in the present form" is here selected, it is again noted that a brief discussion of the potential application to the new frontier of smart roads and real-time decision platforms would add a lot to the manuscript (cf. C5. Management. What about the need to incorporate sensor-based information? is there any possibility to deal with smart roads (cf. 10.3390/electronics8101180 and 10.1016/j.autcon.2018.07.001)).
R1. We further explained the contribution of this study in the conclusion section. In particular, we described how this study's results could be referenced in the process of information connection and data mapping in smart infrastructure technology.
The following sentences are included on page 18, section 5 of the manuscript: “We expect the proposed process in this study ... essential keys of successful BIM for infrastructure ([31,32]). The integrated operation process of the BIM authoring tool–IFC–engineering document proposed in this study can be a good reference.”
Reviewer 3 Report
1. The manuscript is better and can be published, but the authors should improve the conclusions. Please specify the specific 3-4 final remarks resulting from the presented study (in bullets) that may be helpful to the readers.
2. Besides, some figures (4, 7 and 13) have Chinese letters. They should be corrected into English.
3. Section 4 should be presented in another way. Now that's one stream of hard-to-read sentences.
Author Response
C1. The manuscript is better and can be published, but the authors should improve the conclusions. Please specify the specific 3-4 final remarks resulting from the presented study (in bullets) that may be helpful to the readers.
R1. We further explained the contribution of this study in the conclusion section.
The following sentences are included on page 18, section 5 of the manuscript: “We expect the proposed process in this study ... essential keys of successful BIM for infrastructure ([31,32]). The integrated operation process of the BIM authoring tool–IFC–engineering document proposed in this study can be a good reference.”
C2. Besides, some figures (4, 7 and 13) have Chinese letters. They should be corrected into English.
R2. According to the method proposed in this study, document analysis was for documents written in Korean, and we thought it would be desirable to present the results as it is. However, as we judged your point to be more valid, we added English-translated words for the contents in Figures 4, 7, and 13 to enhance the reader's understanding.
C3. Section 4 should be presented in another way. Now that's one stream of hard-to-read sentences.
R3. We divided section 4 into two sub-sections and added Figure 15 to improve the reader's understanding of the information connection contained in Figures 7, 9, and 14.